# Play–Sleep Nexus in Indonesian Preschool Children before and during the COVID-19 Pandemic

**DOI:** 10.3390/ijerph191710695

**Published:** 2022-08-27

**Authors:** Puji Yanti Fauziah, Erma Kusumawardani, Soni Nopembri, Rizki Mulyawan, Indri Hapsari Susilowati, Susiana Nugraha, Sudibyo Alimoeso, Bonardo Prayogo Hasiholan, Lukman Fauzi, Widya Hary Cahyati, Tandiyo Rahayu, Terence Buan Kiong Chua, Michael Yong Hwa Chia

**Affiliations:** 1Nonformal Education Department, Faculty of Education Science, Universitas Negeri Yogyakarta, Sleman 55281, Indonesia; 2Department of Physical Education, Faculty of Sports Science, Universitas Negeri Yogyakarta, Sleman 55281, Indonesia; 3Department of Sports Science, Faculty of Sports Science, Universitas Negeri Yogyakarta, Sleman 55281, Indonesia; 4Department of Occupational Health and Safety, Faculty of Public Health, Universitas Indonesia, Depok 16424, Indonesia; 5Department of Public Health, Faculty of Health Science, Universitas Respati Indonesia, Yogyakarta 55281, Indonesia; 6Indonesian Demographer Association, Depok 16425, Indonesia; 7Directorate of Productive and Older Adult Ages Health, Ministry of Health, Republic of Indonesia, Jakarta 12950, Indonesia; 8Public Health Department, Faculty of Sports Science, Universitas Negeri Semarang, Semarang 50229, Indonesia; 9Physical Education Department, Faculty of Sports Science, Universitas Negeri Semarang, Semarang 50229, Indonesia; 10Physical Education and Sports Science Academic Group, National Institute of Education, Nanyang Technological University, Singapore 639798, Singapore

**Keywords:** play, sleep, preschool children, COVID-19 pandemic

## Abstract

The COVID-19 pandemic has transformed Indonesians’ behaviors and has had an impact on many facets of daily life. People’s lives are becoming increasingly dependent on digital technologies, which is a phenomenon with conflicting effects on people’s health and happiness. This cross-sectional study focused on one such influence, namely, how the shift from the period before to during COVID-19 has affected children’s playtime and sleep duration. As part of a multicenter study, 618 adult caregivers (parents, family members, or babysitters) who visited the kindergarten in question on behalf of preschool children aged 2–5 years (4.04 ± 1.39) were surveyed on the children’s play and sleep habits before and during the COVID-19 pandemic in Indonesia, particularly Java Island (before pandemic, N = 309; during pandemic, N = 309). ANOVA was used for a statistical analysis to describe the difference between groups and within time collections. Significant favorable relationships were found between pre-pandemic and post-pandemic playtime and sleeping time on weekdays, weekends, and averaged weekday-weekend (r = 0.437; 0.180 and 0.321, all *p* < 0.05) were detected. Before the pandemic, children’s playtime (4.11 vs. 3.55 h) and sleep duration (10.92 vs. 10.70 h) were significantly greater on the weekend than on the weekday (*p* < 0.05) but not during the pandemic (playtime: 3.48 vs. 3.45 h and sleep duration: 10.83 vs. 10.80 h; all *p* > 0.05). The COVID-19 pandemic had no impact on sleep duration or playtime in Javanese preschool children. Efforts should be intensified to promote the value of playtime and sleep duration among children in this age range so that the future of Indonesian children’s can be ensured.

## 1. Introduction

One of the ways to encourage children to learn is to play and therefore play becomes a critical part of children’s lives. Play is an activity performed repeatedly to give pleasure and also aids in achieving developmental milestones. Play is also means to reduce or divert stresses caused by the problems in life [1]. While playing, children are engaged in activities that allow them to use their skills, express themselves physically, be creative, and prepare themselves to act and behave maturely [2]. Play is also closely related to children’s physical wellbeing. Children are often engaged in physical activities while participating in organized play and dance in daycare, kindergarten, and primary school, and at the playground or park after school [3]. Play experiences, directly or indirectly, are important for children and have an impact on children’s social and emotional development [4] as well as academic achievement [5].

The importance of play in early childhood has been demonstrated, as some schools use it to increase students’ motivation to learn. Previous studies have shown that the children’s playing environments vary. The types of play can be adapted to children’s interests by considering their social and economic factors. Understanding students’ socioeconomic backgrounds will help schools develop innovative strategies for play activities [6]. Therefore, play—both indoor and outdoor—is seen as a meaningful activity to increase children’s motivation to learn either in school or outside of school.

Humans also need adequate amounts of sleep to maintain optimal health. In fact, sleep patterns alter during the course of one’s life. Sleep also allows cells inside the human body to repair and regenerate. Sleep is understood and characterized by different models, types, or patterns that remain relatively stable at each stage of the lifespan [7]. These characteristics include the sleep schedule and waking time, the sleep rhythm, and the sleep frequency in a day, and sleep satisfaction [2]. Having quality sleep is important for one’s cognitive performance [8,9]. However, many families face difficulties managing children’s sleep habits. Previous research showed that mothers and children have more sleep problems at the ages of 3, 6, and 10 years old [10]. Other research showed that children from low-income families, single-parent families, or with parents with obesity experience shorter sleep durations [11]. Therefore, providing healthy sleep information to caregivers is very important so that good habits can be inculcated in young children.

Effective sleep practices include going to bed early, limiting screen time, and maintaining a regular sleep routine [12]. Approximately one-third of Italian children aged between 1- and 14-years old sleep for less than the minimum recommended hours, and half of them are adolescents. One study showed that the modifiable risk factors, such as limiting screen device use before bedtime and removing screen devices from the bedroom, can help improve sleep quality [13]. There are scant data on the sleep habits of Indonesian children. In ordinary times, the period just before the COVID-19 pandemic, the play–sleep nexus in Indonesian preschool children was not examined.

The COVID-19 pandemic has changed the way children, families, and communities live. Families face multiple challenges such as restrictions placed upon public activities and they must try to adapt to changes in daily living [14]. Since the onset of the pandemic, there is a tendency for preschool children not participate or spend time in organized physical activities or outdoor activities; instead, they spent more time indoors [13]. This has led to an increase in unhealthy lifestyles among preschool children, such as poor quality sleep owing perhaps to an increase in screen time [15,16].

Concerns over the negative influences of excessive screen time on children’s sleep patterns, among other factors, have been raised [17]. In 2019, the World Health Organization (WHO) published global guidelines on physical activities, sedentary behavior (SB), and sleep for preschool children [18], [19]. The early childhood guidelines stipulate that for optimal health, preschool children (aged 2–5 years) should have adequate levels of physical activity, limit sedentary behavior, and have enough sleep every 24 h. Each of these behaviors or their combination(s), when complied with, contribute(s) to the development of optimal levels of physical and mental health in preschool children as it strengthens their immune systems [20].

Various studies demonstrating an association between play and sleep particularly during the COVID-19 pandemic are emerging and have provided different interpretations. For instance, preschool children were sleeping 8% more and were engaged in 16% more moderate to vigorous physical activity, assessed using accelerometers, while parent-reported data showed that these same children were less physically active and had more screen time during the COVID-19 pandemic [21]. Compared to the pre-COVID-19 pandemic, the number of preschool children adhering to the WHO’s recommended hours of sleep generally decreased during the pandemic, and with a significant decrease in compliance to the recommended hours of screen time [22,23].

Meta-analytical research showed that the COVID-19 pandemic has exacerbated the mental health of children aged 5–13 years and brought about negative impacts on prosocial behaviors and caused hyperactivity, with greater effect sizes in European compared to Asian populations [4]. Taking these results at face value, it is critical to implement appropriate strategies in Indonesia that can help increase physical activity; reduce sedentary behavior, including screen time; and ensure that preschool children obtain sufficient sleep during the COVID-19 pandemic to keep longer-term health risks at bay [3].

When the COVID-19 pandemic first emerged, less time was spent on physical activities while screen time and sleep duration increased, but the quality of sleep decreased. Toddlers and preschoolers had limited space to play at home and those living in rural areas were greatly affected by the restrictions imposed. These difficulties were reflected in changes in physical activity levels, screen time, and the quality of sleep. Older children and those living in apartment buildings, where space is more limited compared to rural spaces, experienced even greater impacts with decreased total physical activity and increased screen times. The results of previous studies confirmed that when physical activities decrease, screen time and sleep duration usually increase [13].

For children, screen time and outdoor playtime were closely related to sleep duration and sleep patterns. Reducing screen time and increasing outdoor play time can help improve children’s quality of sleep [24]. An adequate amount of sleep with acceptable quality is crucial for children’s health and development. Even though reduced physical activities and increased screen time have a negative impact on older children’s sleep quality, relatively little information is available on the equivalent impact on children aged below five years. The scarce research examining the relationship between physical activity, outdoor play, and sleep quality in toddlers and preschoolers has produced inconsistent results [25]. It is therefore important to examine the playtime–sleep nexus in preschool children before and during the COVID-19 pandemic so that ‘just-in-time’ interventions to protect children’s health and welfare can be implemented in order to minimize any negative impact on the play–sleep nexus in preschool children that occurred as a result of the COVID-19 pandemic.

## 2. Objectives of the Research

Prior to conducting the full study, pilot data from preliminary research involving caregivers before the COVID-19 pandemic increased our understanding of how play activities might influence children’s quality of sleep [25]. In essence, the pilot results of parent/caregiver interviews showed that the more children became involved in physical play, the easier, longer, and better they slept. However, during the pandemic, the habits of children inevitably changed, with longer engagement in non-physical activities. Nonetheless, these pilot observations require confirmation in a larger-sized sample.

### 2.1. Primary Objective

Therefore, the primary purpose of the present study was to investigate the changes in the lifestyle behaviors of preschool children with a specific focus on two aspects—playtime and sleep—before and during the COVID-19 pandemic.

### 2.2. Secondary Objective

A secondary purpose of the present research was to compare the playtime and quantity of sleep of preschool children against the WHO international guidelines on physical activity and sleep duration for preschool children aged 2–5 years.

## 3. Materials and Methods

### 3.1. Research Design and Institutional Ethics Clearance

The present research was part of a multicenter study that involved urban cities in Asian countries such as Singapore, Malaysia, Taiwan, Thailand, China, India, South Korea, and Japan [26]. However, the present study is focused on data collected in Indonesia, particularly on Java Island. To gather information on kindergartens in the surrounding area, a team from Universitas Negeri Yogyakarta, Universitas Indonesia, and Universitas Negeri Semarang worked together to accomplish the research. Each team from a different institution visited partner kindergartens with permission to survey all adult parent/caregiver of children aged between 2–5 years in order to collect information about the daily lifestyle habits of the children before and during COVID-19 pandemic. The present study was a research collaboration between the cited universities in Java and the National Institute of Education, Nanyang Technological University in Singapore.

The research was approved by the ethics committees of the Institute of Research and Community Service, Universitas Negeri Yogyakarta, Indonesia (540/UN34.21/TU/2019) and that of the Nanyang Technological University, Singapore (NTU IRB-2017-09-036).

### 3.2. Participants

The total data set for this study comprised 618 parents/caregivers with children aged 2–5 years attending multiple kindergartens on Java Island. Parents/caregivers, with informed consent were recruited using a snowball sampling technique and they completed an online questionnaire package on a secure Qualtrics platform. The same sample of parent/caregivers completed the questionnaires before (N = 309) and during (N = 309) the COVID-19 pandemic. The characteristics of the parents/caregivers are summarized in Table 1.

### 3.3. Data Collection

An online questionnaire (Surveillance of digital Media hAbits in earLy chiLdhood Questionnaire, SMALLQ^®^) was completed by parents or caregivers of children aged between 2 and 5 years old where the children’s digital (screen time, type, and purpose) and non-digital media habits (indoor and outdoor play, sleep duration, and quality) on a weekday and weekend were obtained. SMALLQ^®^ comprised three segments: (I) digital media use by parent and child, (II) child’s non-digital behaviors, and (III) child and parent information. SMALLQ^®^ was validated for use with parents of children aged 2–5 years and was adjudged as having an appropriate internal consistency [27]. SMALLQ^®^ was translated into the native Indonesian language by a panel of local experts using an established WHO process and evaluated in pilot trials (visit kindergarten, provide instructions, and administer a questionnaire to a small group of test participants) that included cognitive interviews, with relevant adjustments to the questionnaire produced prior to the main study. Online informed consent was obtained at the beginning of the questionnaire.

The inclusion criteria of participants for our study were parents of preschool children aged 2 to 5 years old. The survey was hosted on a secured online survey platform called Qualtrics^®^XM (https://www.qualtrics.com/) (accessed on 1 January 2020) and completed by parents or caregivers who met the inclusion criteria.

### 3.4. Data Analysis

Data were analyzed using SPSS version 23.0 statistical software. The analyses included means comparison, independent *t*-test, and descriptive statistics of variables of interest before and during the COVID-19 pandemic. The analyses focused on comparing and correlating children’s playtime and sleep duration before and during the COVID-19 pandemic on weekdays and weekends.

## 4. Results

### 4.1. Caregiver Characteristics

The caregivers in this study were mothers, fathers, grandmothers, grandfathers, and household assistants, with a majority of mothers completing the questionnaire. In addition, the category of education status varied widely, ranging from those with no formal education to those with a master’s degree. The majority of parents/caregivers attained a bachelor’s degree or had a high school education qualification (Table 1). 

### 4.2. Playtime and Sleep Duration

Table 2 presents information about weekday and weekend playtime and sleep duration derived from the SMALLQ^®^ questionnaire before and during the COVID-19 pandemic. Before the COVID-19 pandemic, children’s playtime was significantly higher on the weekend than on the weekdays (4.11 vs. 3.55 h, *p* < 0.05 or 15.7% higher) while sleep duration was the same on the weekday and on the weekend (10.70 vs. 10.92 h, *p* > 0.05). During the COVID-19 pandemic, both weekday and weekend playtimes and sleep durations were not significantly different from each other (*p* > 0.05). This is shown in Figure 1.

### 4.3. Playtime–Sleep Nexus

The relationships between playtime and sleep duration were explored using bi-variate correlations. Longer playtimes were associated with a longer sleep duration. Significant correlation coefficients (*p* < 0.05) between playtime and sleep duration on weekdays, weekend, and weekly (mean of weekday and weekend) were 0.437, 0.180, and 0.321, respectively. This is depicted in Table 3.

## 5. Discussion

### 5.1. COVID-19 Pandemic as a Great Disruptor

The COVID-19 pandemic is a great disruptor and has impacted children’s lives. Children who used to spend weekdays in school and playing outdoors during school breaks and after school were compelled to stay home and accomplish mainly sedentary activities, including online learning, from the home. When restrictions were first imposed, children lost opportunities to participate in organized physical activities, spend time playing outdoors, and playing with friends. In addition, sport and recreational activities were largely forbidden or restricted, and nearly all playgrounds were closed, with children reminded to maintain social distancing at all times [28]. While these restrictions were implemented to limit infections for the greater good of the population (e.g., to avoid hospitals from being overstretched with admissions), research has confirmed the benefits of physical activities for physical and mental health, psychosocial skills, academic achievement, and a stronger immune system, which are critical during the COVID-19 pandemic [29].

### 5.2. Playtime in Preschool Children before and during the COVID-19 Pandemic

The primary purpose of the present study was to investigate the playtime–sleep duration nexus among Indonesian preschool children aged 2–5 years before and during the COVID-19 pandemic. Before the pandemic, playtime over the weekend was 15.7% longer than on weekdays. The weekday and weekend playtimes were similar in duration during the pandemic. Additionally, playtime on the weekdays and weekend before and during the pandemic still met the WHO guidelines of 3 h of play daily for preschool children [30].

Elsewhere, similar observations of greater weekend than weekday physical activity were observed, as there is more time over the weekend and preschool children are free from school-based routines, where play is limited to recess and after school on weekdays. For instance, Faulkner et al. reported that older children aged 10 years played more outdoors on weekends than on weekdays and that the play duration was correlated with the amount of moderate-to-vigorous physical activity [31].

Before the COVID-19 pandemic, plausible reasons for a decreased playtime for children include the use of gadgets, parents working long hours to provide for their family, an increased learning content in preschool, and an increased focus on structured academic activities, within and outside of schooling hours (Zamorano, Abad, Hernández, Herrera, and de la Fuente, 2019 [32]).

In the present study, during the COVID-19 pandemic, weekday and weekend play was reduced, probably because of the curtailment of movement and group size restrictions because of COVID-19 restrictive measures. Moreover, vaccinations for children aged 2–5 years were not yet available. Equivalent parent-reported data revealed similar trends whereby playtime or physical activity were decreased because of the COVID-19 pandemic [33]. In the present study, the sleep durations remained unchanged before and during the COVID-19 pandemic. Additionally, the preschool children had 10.81 h of sleep, which is within the 10–13 h of sleep recommended by WHO.

In the systematic review of studies focused on the impact of COVID-19 on movement behaviors, the majority of the research showed that children and adolescents under COVID-19 restrictions were likely to be less physically active, have greater screen times, and have longer sleep durations than before the COVID-19 lockdown [33]. It should be noted that not all the studies show similar results. While parent reports suggest that preschool children were less physically active, slept less, and spent more time being sedentary, Ng et al. (2021) reported that preschool children increased MVPA by 16%, spent 8% more time sleeping, and spent 9% less time engaging in sedentary behaviors—assessed using an accelerometer—during the COVID-19 pandemic than before [21].

The roles of play in an educational setting are important. As a unique means of interaction among children, play is a natural means for child development and learning. Play contributes to balance in the lives of children, which at the same time represents an activity, an adventure, an experience, and a means of communicative interaction with others that depends on the free time given to children under parents’ supervision. Play is also a part of a comprehensive educational process, greatly needed to help develop children’s bodies, intelligence, and social ability [34]. It is an activity commonly used to develop motor skills, visual-motor coordination, motor (fine and gross) coordination, and social interactions among boys and girls [32]. In addition, play increases executive functions, language ability, mathematic skills (dealing with numbers and spatial concepts), social development, relationship with peers, physical development and health, and emotional health while reducing anxiety and stress [32]. While many of the cited benefits of play are reported in older children, the importance of play among preschool children cannot be over-emphasized as it may serve as the most effective therapy to mitigate the inimical impacts of the COVID-19 pandemic.

### 5.3. Sleep Duration in Preschool Children before and during the COVID-19 Pandemic

Sleep is undeniably important for young children, particularly because growth hormones needed for physical growth is released when children sleep [35]. Several studies elsewhere showed that during the COVID-19 pandemic children became less physically active, displayed more sedentary behaviors, preferred screen-based recreational activities, and spent more time sleeping compared to pre-pandemic periods [33,36,37].

In the present study, the weekend sleep duration was significantly greater (8% more) before the COVID-19 pandemic than during the pandemic. Some of this weekend sleep could plausibly be catch-up sleep as weekday sleep was less than weekend sleep. However, during the COVID-19 pandemic, weekday and weekend sleep durations in preschool children were not significantly different, perhaps because school adjusted to online learning with no need to leave the home to travel to school. Children’s sleep before and during the COVID-19 pandemic was at the lower end of the WHO sleep recommendations of 10–13 h daily.

Given how sleep affects children’s health and development, the quality of sleep is critical for children. It appears that few Indonesian caregivers know that reducing physical activities—including playtime—and increasing screen time, affect children’s quality of sleep [25,38,39]. A shorter quality sleep is associated with a higher chance of adiposity, emotional dysregulation, growth disorder, screen time, and risks of injury [40]. In short, lifestyles determine children’s well-being [41], and caregiver and family settings are important avenues for inculcating and socializing healthy lifestyle habits for preschool children.

Childhood is a critical growth period, and some habits beginning in this period will continue as one grows into adulthood. For children, living a healthy lifestyle brings about positive benefits to their physical health and cognitive performance [42]. Learning quality sleep habits when young has relevant and comprehensive effects on children’s development [43]. Other research has shown that proper and safe sleep allows children to not only develop rhythms and habits that will be retained for the rest of their lives but also to grow well both physically and cognitively. Contrarily, children who lack sleep usually show increased levels of anxiety and aggressiveness as well as poor cognitive performance and memory. Chronic sleep deficiency can lead to sleep disorders that disrupt the normal functioning of the endocrine system and potentially lead to eating disorders, childhood obesity, sleep apnea, and hyperactivity [44]. Therefore, it is important to continue to monitor sleep duration in preschool children throughout the COVID-19 pandemic and beyond.

### 5.4. Playtime–Sleep Duration Nexus and Compliance to WHO Guidelines

A secondary purpose of the present study was to investigate the playtime–sleep duration nexus in preschool children before and during the COVID-19 pandemic. The results of the present research showed small to moderate positive and significant correlations between playtime and sleep duration only before the COVID-19 pandemic. These findings are consistent with what others have found before the COVID-19 pandemic [11]. For instance, one study reported that shorter sleep durations among children were associated with a decrease in physical activity while another study showed that total playtime was associated with nighttime sleep after 9 p.m. [39]. During the COVID-19 pandemic, the associations between playtime and sleep duration did not attain statistical significance. In the present study, weekly playtime was reduced while weekly sleep remained unchanged. These observations are in agreement with parent-reported data by researchers in another study who studied physical activity and sleep among preschoolers living in Hong Kong, but is also contrasted with accelerometer-derived physical activity and sleep data in the same group of 25 preschoolers [21]. The similarities and differences between the two studies could plausibly be explained by the different contexts between the countries before and during the COVID-19 pandemic, and the use of questionnaires versus accelerometers to obtain physical activity and sleep data. Therefore, further research is recommended to better explicate the playtime–quantity sleep nexus.

In terms of playtime and sleep, the WHO guidelines stipulate that children in this age group should have at least 180 min of physical activity per day, inclusive of 60 min of activity of moderate to vigorous intensity, and that preschoolers should have at least 10–13 h of regular quality sleep [18]. The parent/caregiver-reported mean playtime and mean sleep duration data in the present study met the WHO guidelines before and during the COVID-19 pandemic. Notwithstanding these favorable findings, the continued and regular surveillance of the digital and non-digital behaviors of preschool children are necessary to safeguard the holistic development of Indonesian children in early childhood.

### 5.5. Strengths and Limitations

The present study investigated the playtime–sleep duration nexus in a modest-sized sample of preschool children living in Java Island before and during the COVID-19 pandemic using a content-validated and reliability-tested parent-reported online SMALLQ^®^ questionnaire. The limitations include social desirability and recall biases, because nearly 80% of the caregivers who completed the online SMALLQ^®^ questionnaire had graduate degrees (at least polytechnic). However, some of these potential biases were mitigated by keeping the parent-report anonymous and all recall of child play and sleep durations limited to the last seven days. As the research is cross-sectional and the play–sleep associations were determined by bi-variate correlations, no cause-and-effect relationship can be established. Additionally, as Indonesia is a very large country with many different cultures and contexts, the present findings cannot be extrapolated beyond Java Island. More research using other established objective methods and/or in combination with subjective methods for assessing the lifestyle behaviors of preschool children is recommended to affirm the present findings. This research can offer a preliminary study of the development of preschoolers’ physical literacy in Indonesia.

## 6. Conclusions

Before the COVID-19 pandemic, the playtime and sleep duration among preschool children were positively associated; a longer playtime was associated with a longer and better sleep duration, and both playtime and sleep duration were significantly higher on weekends than on weekdays in preschoolers. These associations were nullified during the COVID-19 pandemic. The playtime and sleep duration before and during the COVID-19 pandemic complied with the WHO physical activity and sleep guidelines for this sample of Javanese preschoolers. The continued surveillance of playtime and sleep among preschoolers using a combination of subjective and objective methods is recommended during and beyond the COVID-19 pandemic to safeguard the health and wellbeing of preschoolers living in Indonesia.

## Figures and Tables

**Figure 1 ijerph-19-10695-f001:**
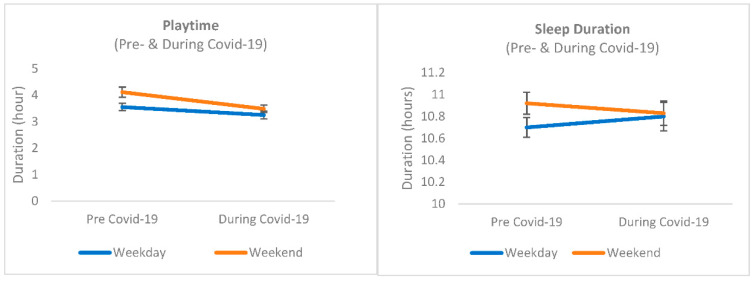
Comparison of playtime and sleep duration before and during COVID-19.

**Table 1 ijerph-19-10695-t001:** Participants’ Characteristics.

Parameter	Pre-COVID-19 Groups	During COVID-19 Groups	All Groups	
			Total	%
**Relationship to child:**				
Mother	271	205	476	77.02
Father	27	66	93	15.05
Grandmother	3	17	20	3.24
Grandfather	2	3	5	0.81
Legal guardian	6	18	24	3.88
**Educational Status**				
No formal education	1	0	1	0.16
Primary School	6	7	13	2.10
Secondary School	109	4	113	18.28
Academy/Polytechnic	34	75	109	17.64
Bachelor’s Degree (S1)	132	182	314	50.81
Master’s (S2)/Doctoral Degree (S3)	27	41	68	11.00
**Number of Children (n = 618)** Child age (years on average)	3.84	4.22	4.045	100

**Table 2 ijerph-19-10695-t002:** Playtime and sleep duration of preschool children before and during the COVID-19 pandemic.

	Pre COVID-19 (Time)	During COVID-19 (Time)	Mean Differences	t	Sig.
**Playtime (hours)**					
Weekday	3.55 ± 0.14 ^a^	3.25 ± 0.19	0.304	1.268	0.205
Weekend	4.11 ± 0.15 ^b^	3.48 ± 0.14	0.635	3.166	0.002
Weekly	3.83 ± 0.14	3.36 ± 0.15	0.470	2.284	0.023
**Sleep duration (hours)**					
Weekday	10.70 ± 0.09	10.80 ± 0.13	−0.093	0.548	0.584
Weekend	10.92 ^a^ ± 0.10	10.83 ± 0.11	0.092	0.591	0.555
Weekly	10.81 ± 0.09	10.81 ± 0.12	−0.000	0.003	0.998

^a^ Denotes value is significantly different between weekdays and weekend pre-COVID-19. ^b^ Denotes value is significantly different between pre-COVID-19 and during COVID-19.

**Table 3 ijerph-19-10695-t003:** Correlations between playtime and sleep duration before and during COVID-19 pandemic.

Association between Playtime and Sleep Duration	Correlation	
	Pearson	Sig.
Weekday: Playtime vs. Sleep duration	0.437	*p* < 0.05
Weekend: Playtime vs. Sleep duration	0.180	*p* < 0.05
Weekly: Playtime vs. Sleep duration	0.321	*p* < 0.05

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
