# Peer review of "Play–Sleep Nexus in Indonesian Preschool Children before and during the COVID-19 Pandemic"

_ijerph, 2022, doi:10.3390/ijerph191710695_

Round 1

Reviewer 1 Report (Previous Reviewer 2)

1.     This is an improved version of the manuscript.

2.     The title is current and interesting.

3.     The abstract appropriately conveys the content of this text.

4.     The manuscript contributes to the analysis of the play-sleep nexus in Indonesian preschool children.

5.     This is a good topic for educational references in teaching preschool children during the COVID-19 pandemic.

Author Response

Thank you so much for your positive comments to our article. 
All authors appreciate it! 

Warm regards.

Reviewer 2 Report (Previous Reviewer 3)

I think the article can be accepted

Author Response

Thank you so much for your positive comments to our article. 
All authors appreciate it! 

Warm regards.

Reviewer 3 Report (Previous Reviewer 4)

Manuscript ID: ijerph-1862796

Title: Play-sleep nexus in Indonesian preschool children before and during the COVID-19 pandemic

Journal: International Journal of Environmental Research and Public Health

General comment

Overall, Authors have adequately addressed my previous concerns. However, I would have other suggestions.

Specific comments

Materials and methods

1.     Line 173. Authors should replace “Subjects” with “Participants”.

2.     Line 186. Authors should quote the trials they are referring to.

Results

1.     Lines 197-199. I am still convinced that those data (i.e., 80% of the caregivers with higher education levels) may actually highlight a reduced generalizability of the results. Authors should take into account this comment within the limitations paragraph.

2.     Table 2. Other than the p-value, Authors should add the complete statistics.

3.     Table 3. Authors should clarify that this table (that should be quoted within the main text) only shows the data observed before the Covid-19 pandemic.

Author Response

Thank you so much for your positive comments on our article. 
All authors appreciate it! 

And also we sent you some revisions based on your additional comments. Please see the attachment.

Warm regards.

Reviewer 4 Report (New Reviewer)

The presented study did not bring anything new to the knowledge related to area of expertise.

It has got a few serious flaws related to the description of introduction, methodology and presenting results. In my opinion also used statistics is poor.

Introduction: Authors have to present latest outstanding original articles related to the topic and then write why their study is new and important.

Materials and Methods: You have to provide detailed information about study. Precise places of data collecting on Java Island, detailed inclusion and exclusion criteria of participants, and detailed information of survey distribution (You can't write only "online questionnaire was completed by parents or caregivers"). The online survey can be delivered to the respondent using a plenty of tools. Please provide precise information.

Furthermore, Authors have to provide information which University or Institution has been responsible for this study on Java Island and how the University/Institution and kindergartens cooperate?

Statistics and Results: I think that Authors should use more reliable test than means comparison, independent t-test, and descriptive statistics of variables. I suggest to consider linear regression analysis.

After a new statistic is made, the entire Results section should be rewritten.

Author Response

Thank you so much for your comments on our article. 
All authors appreciate it! 

I sent you some revisions based on your additional comments. Please see the attachment.

Warm regards.

This manuscript is a resubmission of an earlier submission. The following is a list of the peer review reports and author responses from that submission.

Round 1

Reviewer 1 Report

The article deals with a subject of great relevance for scientific knowledge in the field of children's right to play and sleep. It is well written and structured. It presents a methodology that is well-oriented and adequate for the study. The discussion of the results is based on the theoretical framework and on the empirical data. In my opinion, the conclusion should be more enriched with content, namely associating the Convention on the Rights of the Child with the right to play and the quality of sleep, referring to the importance of the recognition of the child's citizenship in contemporary society. 

Reviewer 2 Report

1. This is a concise manuscript.

2. The authors make a contribution to the research in the area of play-sleep nexus in preschool children before and during the COVID-19 pandemic.

3. The methods are generally appropriate.

4. The topics discussed to meet the needs of the current COVID-19 pandemic in the preschool education field.

5. The conclusion could strengthen by adding specific statements about the contributions of this study.

Reviewer 3 Report

Play-sleep nexus in Indonesian preschool children before and

during the COVID-19 pandemic (ijerph-1795688)

(Review)

Main message of the article

The article “Play-sleep nexus in Indonesian preschool children before and during the COVID-19 pandemic” examines the impact of COVID-19 pandemic on preschool children’s playtime and sleep quality. The authors found significant correlations between playtime and quality of sleep. Furthermore, patterns of playtime and sleep quality were assessed in weekdays and weekends before and during COVID-19 pandemic, and levels seemed to meet the WHO guidelines.

General Judgment Comments

The article explores an interesting topic in relation to play and sleep quality during COVID-19 pandemic. Nevertheless, the article is not very clear to follow, and the logical order of statements is often chaotic for the reader. The Abstract does not provide adequate information to summarize and properly understand the study. For instance, the aims of the research, methods, and statistical analysis are not clarified. The conclusion of the abstract seems quite generic. The Methods and the Study Design need more information to be clear. Data analysis is not clearly presented, and it does not always address the reported aims of the research. For instance, to investigate changes in lifestyle behaviors before and during the COVID-19 pandemic, it would have been better to compare pre- and during COVID-19 scores for the variables of interest. By not doing so, we do not know whether variations between scores reported pre and during the pandemic are statistically significant or if they are just random fluctuations. In the manuscript, results are not presented following the standard format and no effect size is provided. Ultimately, no information regarding informed consent and data treatment is provided.

For all these reasons, and for the specific Major and Minor Issues identified below, I would recommend the manuscript for Rejection.

Major Issues

-       In the Abstract, the passage from the first sentence (related to Covid-19) to the second (related to digital technology dependence) is not clear.

-       Abstract: What do the authors mean for “this dependence has mixed impact of health and wellbeing”?

-       Abstract: the sentence “One such impact-related to children’s playtime and quality sleep duration, is the focus of the present study” is quite vague. What is the focus of the present study?

-       The aims of the study, the adopted methods, and the statistical approach are not clear in the abstract.

-       In the Abstract, please clarify the number of children and add also the mean and the standard deviation for the children’s age. Also, please report the age information for both samples of children (before and during COVID-19)

-       In the Abstract and in the main text, please report results following the standard format (including the specific p-values).

-       In the Abstract, please clarify how caregivers were surveyed and, if adopted, which questionnaires were used.

-       From the Abstract, it is not clear whether the study was longitudinal or cross-sectional.

-       Abstract: “Significant positive correlations between the playtime and quality sleep time on the weekday, 31 weekend, and averaged weekday-weekend (r=0.437; 0.180 & 0.321, all p<0.05) were detected”. Is this sentence referring to the period before or during the Covid-19 pandemic? Please clarify.

-       Abstract: “[…] quality sleep (10.92 vs.10.70) […]” what do these scores represent? Please clarify.

-       Have the authors used a statistical method to control the significance level for multiple tests (e.g., Bonferroni correction)? By conducing all these tests, there is an increased risk of type-I error.

-       “It appears that despite restrictive measures imposed because of the COVID-19 pandemic, Javanese preschool children were able to meet the World Health Organisation guidelines for playtime and quality sleep”. Please clarify. This conclusion is quite generic, and it does not seem to be connected with the analysis conducted in the paper. Also, did the authors conduct any comparison of sleep quality and playtime before and during COVIS-19? It would have been useful to know whether levels of sleep quality and playtime before and after the pandemic were significantly different.

-       “Efforts should be sustained to continue to promote the importance of playtime and quality sleep among caregivers of children in this age group so that the future good health of Indonesian children can be secured”. Please clarify this sentence and how it relates to the study’s results. Even though it is desirable to continue to promote the importance of playtime and quality sleep, it is not clear the way in which this lesson was learnt from the results of the current study.

-       Introduction: “Play is a means to reduce or divert stresses caused by the problems in life”. Please clarify, do children divert stresses caused by the problems in life when playing?

-       More references are needed to support the statements made by the authors in the Introduction.

-       Overall, the Introduction needs more work and structure.

-       Introduction: what do the sentences “Sleep also allows cells inside the human body to repair”? And “Sleep is understood and characterised by different models, types, or patterns that remain relatively stable period”? Please clarify and add references.

-       Lines 86-87 of the Introduction: please clarify the link between spending time indoor and poorer sleep quality.

-       The Introduction section is quite lengthy.

-       The Primary objective requires a direct comparison of scores in playtime and sleep quality documented pre- and during COVID-19.

-       Line 158-159: “However, the present study is focused on data collected in Indonesia, particularly on Java Island”. This should be clarified in the Abstract too.

-       It is not clear whether the same sample of caregivers was interviewed pre and during COVID-19 or if the study relies on two different samples for the two phases.

-       Please provide demographic information for the children.

-       Please provide the psychometric properties of the questionnaires.

-       Please clarify the analytical approach. Also, in the Abstract, the authors reported results from correlation tests, which do not appear to be included in the Data Analysis paragraph. Please also clarify how data were treated (e.g., were there any exclusion criteria in the sample? Were there any participant with missing data?).

-       Was any statistical correction adopted to control for type-I errors?

-       What was the effect size of the significant results?

-       In the results section, results are not reported in the standard format.

-       Figures would help the reader to understand the results.

-       No information regarding informed consent and data treatment is provided.

-       Throughout the manuscript, it seems that the authors have assessed sleep quantity more than sleep quality. Please clarify

Minor Issues

-       Abstract: “The COVID-19 pandemic has affected many aspects of life and changed affected the behaviours of Indonesians.” Please remove the second “affected”.

-       Abstract: “[…] children’s playtime (4.11 vs 3.55 hrs) […]”. Please clarify: hours a day, a week, etc?

Final comments

I would recommend for Rejection.

Reviewer 4 Report

Manuscript ID: ijerph-1795688

Title: Play-sleep nexus in Indonesian preschool children before and during the COVID-19 pandemic

Journal: International Journal of Environmental Research and Public Health

Abstract

1.     Lines 26-27. Authors wrote “One such impact-related to 26 children’s playtime and quality sleep duration, is the focus of the present study”. I believe that they should rephrase the aim of the study.

2.     Lines 31-32. Authors wrote “Significant positive correlations between the playtime and quality sleep time on the weekday, weekend, and averaged weekday-weekend (r=0.437; 0.180 & 0.321, all p<0.05) were detected”. Authors should specify the temporal reference of their statement, i.e., are they referring to the pre-pandemic period, the pandemic period or else?

Introduction

1.     Lines 65-67. Actually, sleep changes across the life span (e.g., https://pubmed.ncbi.nlm.nih.gov/33823052/).  

2.     Lines 71-73. Authors wrote “Other research showed children from low-income families, single-parent families, or with parents with obesity, experience shorter sleep duration”. Authors should quote the study they are referring to.

3.     Line 78. Authors should replace “and14” with “and 14”.

4.     Line 79. The citation style (i.e., “Brambilla et al., 2017”) is not numerical.

5.     Line 109. Authors should quote the meta-analysis they are referring to.

6.     Lines 133-135. Authors should summarize the main findings of the studies they are referring to.

Materials and Methods

1.     Line 163. At the begging of the sentence, the number should be written in letters.

2.     Line 168. Table 1 is not reported.

3.     Lines 178-181. Did Authors use an Indonesian validated version of the questionnaire?

Results

1.     Lines 190-192. I believe that those data (i.e., 80% of the caregivers with higher education levels) may actually highlight a reduced generalizability of the results.

2.     Lines 198-199. Authors wrote “while quality sleep was the same on the weekday and the weekend (10.70 vs 10.92 hrs, p>0.05)”. Instead of sleep quality, Authors should refer to sleep quantity.

3.     Line 211. Within Table 2, Authors should add the complete statistics of the weekday-weekend comparison as well as the pre-Covid 19 vs. during-Covid 19.

4.     Lines 215-216. Authors should add a Table with all the correlation values, both significant and non-significant.

5.     Line 216. Authors should replace “longer quality sleep” with “longer sleep duration”.

Discussion

1.     Line 259. Authors should replace “quality sleep” with “sleep quantity”.

2.     Line 352. Authors should expand the description of these limitations.